

# Simulating landfast sea ice breakage due to ocean eddies using a discrete element model

Rigoberto Moncada[1,2], Mukund Gupta[1,3], Jacinto Ulloa[1,4], Andrew F. Thompson[1], and Jose Andrade[1]

[1]California Institute of Technology, 1200 E. California Blvd, Pasadena, CA 91125
[2]Lawrence Livermore National Laboratory
[3]Delft University of Technology
[4]University of Michigan

**Correspondence:** Jose Andrade (jandrade@caltech.edu)

**Abstract.** Marginal ice zones are influenced by energetic oceanic motions over a range of scales, including forcing due to surface waves and (sub-)mesoscale eddies. While the role of waves in breaking sea ice has been well recognized, the influence of ocean eddies in the fracturing process remains less explored. This work considers simulations of a landfast sea ice pack represented by a bonded Discrete Element Model (LS-DEM-BPM) and forced by eddying ocean currents generated by a quasi-

geostrophic model. These experiments reveal that ocean eddies can generate realistic fracture patterns and floe size distributions (FSDs). For the same amount of eddy kinetic energy, ocean currents with a larger characteristic eddy size penetrate deeper into the pack and fracture more floes. This creates floe distributions with a slightly higher FSD slope as compared to forcing by smaller eddy length scales. On the other hand, stronger bonds between the DEM elements lead to less breakage and a notably shallower FSD. These results are qualitatively consistent with an analytical model of the fracturing process, which provides

an upper limit to the expected breakage area. These insights may help formulate more comprehensive parameterizations of breakage within coarse and continuum-based sea ice models.

## 1 Introduction

Thin layers of frozen surface sea water form sea ice during winter in the polar oceans. In its most consolidated form, this floating ice forms a contiguous pack with anchor points on surrounding land masses (Hwang and Wang, 2022), commonly referred to

as landfast sea ice. This sea ice pack is prone to breakage due to dynamical forcings from the ocean and the atmosphere, particularly in the summer, when melt weakens its structural integrity (Lin et al., 2022; Lei et al., 2020). Individual fragments resulting from these breakage events, called floes, interact with ocean turbulence, disperse into the open ocean, and melt more easily than a consolidated sea ice pack (Horvat et al., 2016; Manucharyan and Thompson, 2017; Gupta and Thompson, 2022; Gupta et al., 2024). Over the last few decades, the average thickness of Arctic sea ice has declined by approximately one half

(Gascard et al., 2019), resulting in a sea ice cover that is likely more vulnerable to breakage events (Barber et al., 2018; Asplin et al., 2014), notably as Arctic cyclones have increased in intensity and duration in recent years (Zhang et al., 2023).

Sea ice is a heterogeneous material subject to a complex set of forces. The physical properties of sea ice are influenced by factors such as varying thickness (tens of centimeters to tens of meters), melt ponds, impurities from minerals carried by the



ocean and atmosphere, inhomogeneous snow cover, cracks, and biological growth (Lund-Hansen et al., 2024; Skatulla et al.,
2022; Golden, 2001). The mechanical forcing scales range from millimeters (Sammonds et al., 2017; Cole, 2001) to kilometers
(King et al., 2018), including winds, waves, ocean currents, and inter-floe collisions. The formation of cracks, driven by such
forcings, is counteracted by freezing processes occurring over hourly to daily time scales. Studying the time evolution of the
observed floe size distribution (FSD) provides insights into these sea ice breakage processes, but it remains challenging to
untangle the role of different forcings (Lopez-Acosta et al., 2019). Simulating the fracture behavior of sea ice packs is thus
essential for understanding how sea ice will evolve in the coming years.

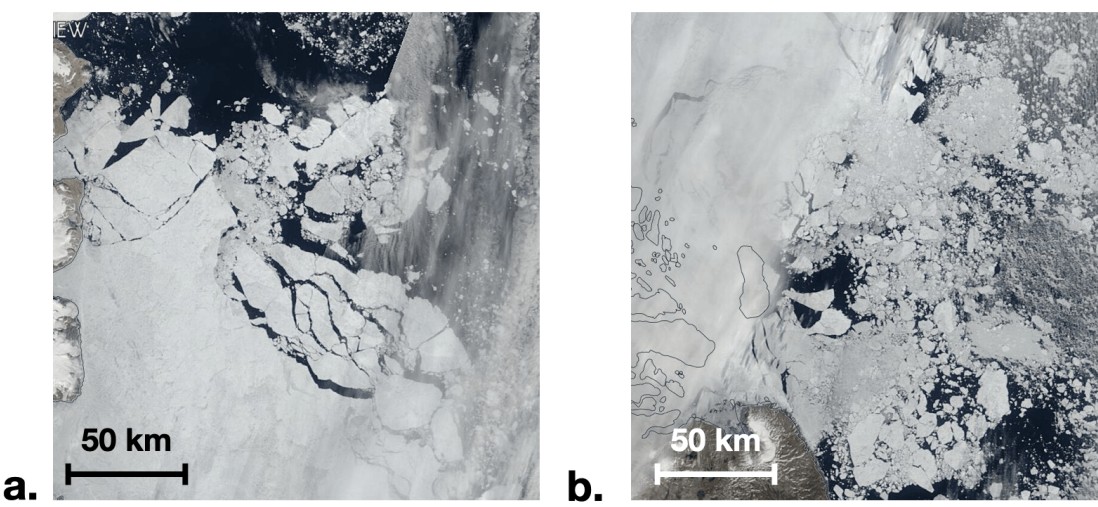

**Figure 1.** Satellite images of the marginal ice zone obtained by NASA MODIS on August 1, 2023, within the Fram Strait region of the Arctic.
(a) Formation of large individual floes (tens of kilometers) as sea ice detaches from the main pack (northwest: N 81.3°, W 17.2°; northeast:
N 79.6°, W 2.4°; southwest: N 79.6°, W 21.2°; and southeast: N 77.8°, W 8.4°). (b) A more fractured marginal ice zone, where individual
floes are less easily distinguishable (northwest: N 79.2°, W 21.8°; northeast: N 77.3°, W 8.6°; southwest: N 77.6°, W 26.0°; and southeast:
N 76.3°, W 14.1°). This work explores the role of ocean eddies in the formation of distinct sea ice patterns, such as the ones appearing in
these images.

Ocean waves are critical to the formation of individual floes at the boundary between the open ocean and the sea ice
pack, namely the marginal ice zone (MIZ) (Toyota et al., 2016; Langhorne et al., 1998). Waves induce off-plane vertical
displacements and bending moments that break sea ice through fatigue and tidal surges (Squire et al., 1995; Langhorne et al.,
1998; Li et al., 2021). The propagation of surface waves and swell underneath sea ice is arrested by ice-ocean friction, energy
loss due to ice breakage, and wave scattering, which is notably dependent on sea ice concentration, thickness, and FSD (Montiel
and Squire, 2017). In landfast ice, waves tend to penetrate up to about 0.8 to 4.5 km into the pack (Kovalev et al., 2020).
Modeling surface wave-sea ice interactions has been developed using both continuum (Roach et al., 2019; Yang et al., 2024)
and discrete element (Herman, 2017; Zhai et al., 2024) approaches, capturing the effects of wave scattering, wave-induced
breakage, and coupled atmosphere, ocean, and sea ice response.



Fractures that occur deeper in the sea ice pack are more strongly influenced by a combination of winds and ocean currents, which act at kilometer-level spatial scales, than waves that affect smaller scales ranging from meters to hundreds of meters (Willmes et al., 2023; Kovalev et al., 2020). These forcings impose shear and tension within the sea ice, leading to the formation of cracks. Winds are the primary drivers of sea ice drift (Spreen et al., 2011; Toyota et al., 2016), and storm events have been observed to trigger large-scale leads and modulate crack propagation paths (NSIDC, 2024, 2013). Winds generally

influence crack formation over larger length scales but shorter periods (weeks) as compared to ocean currents, and are often the primary trigger for fast ice breaking away from its borders (Willmes et al., 2023). Ocean currents, like winds, influence the ice at kilometer-level scales but typically have longer-lasting effects, manifesting over weeks to months. For instance, wind-driven currents cause extensive fracturing in the Beaufort Gyre (NSIDC, 2013). In regions such as the Fram Strait or the Kara Sea, strong currents often play the dominant role in fast-ice breakage patterns.

Instabilities in ocean currents tend to generate coherent features with scales ranging from 1–100 km, termed eddies, which also impact sea ice. Ocean eddies can accelerate sea ice melt by assisting in the transport of warm waters towards the center of individual floes and dispersing floes into the open ocean, where they are more vulnerable to breakage by waves (Gupta et al., 2020, 2024). Observations from satellite imagery also show that eddies within the MIZ strongly influence the fracture process (NSIDC, 2013). In turn, changes in sea ice geometry affect atmospheric and oceanic currents (Watkins et al., 2023;

Willmes et al., 2023), leading to a feedback loop. For instance, sea ice removal results in enhanced wave and eddy activity (Armitage et al., 2020), which in turn favors further breakage and the melting of sea ice. Despite these observations, the role of in-plane ocean currents in sea ice fracture has been less studied than that of waves or winds.

Global climate models typically use a continuum approximation to represent sea ice, which does not accurately capture its granular physics. Alternatively, Discrete Element Methods (DEMs) can be used for studying the behavior of interacting floes

(Herman, 2011, 2013, 2017; Bateman et al., 2019) and the resulting fracture processes (Turner et al., 2022; Åström et al., 2024; Lilja et al., 2021; Brenner and Horvat, 2024). In this framework, a floe is represented by a single element with simple or complex shapes (Moncada et al., 2023; Manucharyan and Montemuro, 2022), or a collection of elements bonded together to form irregular shapes (Herman, 2016; West et al., 2021). Past works have mostly considered the fracturing response of sea ice floes under relatively simple oceanic or atmospheric forcings, where winds and currents do not vary strongly in space

and are decoupled from the floe dynamics. On the other hand, the interactions between floes and ocean turbulence have been investigated in a two-way coupled model (Gupta and Thompson, 2022; Gupta et al., 2024), but without considering breakage.

Here, we focus on isolating and illustrating the role of ocean eddies in the fracture of multi-kilometer landfast sea ice. We consider a bonded particle model (BPM) forced by currents generated from a two-layer quasi-geostrophic model of the ocean. We hypothesize that characteristic eddy sizes modulate the collapse rate, breakage mode and the evolution of the pack. These

breakage modes include conditions where sea ice is broken into floes with sizes comparable to the eddy length scale or those where the ice pack is broken into a slurry, with floes much smaller than the eddy length scale (Figs. 1a and b, respectively). We explore how these differences depend on the characteristic length and velocity of ocean eddies and the ice pack's critical strength.



This paper is organized as follows. Sec. 2 details the BPM framework to simulate fast ice and its breakage. Secs. 3 and 4 present theory regarding different breakage modes or behaviors, related to the spatial distribution of ocean velocity, along with validation from idealized experiments in the BPM. Sec. 5 shows the results of simulations with eddying currents, and Sec. 6 provides a discussion and conclusions on the impact of eddies on sea ice breakage, as well as opportunities for future work.

## 2   The bonded particle method for sea ice

We adopt a discrete element method based on LS-DEM (Kawamoto et al., 2016) and its extension to bonded particles (LS-DEM-BPM) (Harmon et al., 2021). While this method can handle arbitrarily shaped elements, we choose circular disks to keep the computational cost tractable. The disks are arranged in a hexagonal mesh with a maximum achievable packing efficiency of 0.91. The radius of each element is fixed to 500 m, chosen to bound the total number of grains due to computational resources. This size is also similar in resolution to the fluid velocity grid, which is 1 km × 1 km. Each element is connected with up to six of its neighbors using bonds, which behave like cylindrical beams undergoing normal forces (compression or tension), shear forces, and torques. A given pair of elements is connected via one bond at most, for simplicity. For this study, we assume that the only external driving force consists of ocean currents, which drag the elements laterally at their base. The immersion of sea ice within water is not explicitly represented.

In the general context of LS-DEM, a detailed description of inter-particle forces and torques due to collisions and bonds is provided in Harmon et al. (2021). Here, we provide a summary, focusing on how floes respond to the external drag forces. Fig. 2a shows an arbitrary and isolated pair of bonded discrete elements labeled $i$ and $j$, subject to the oceanic drag forces $\mathbf{F}_i^{\mathrm{drag}}$ and $\mathbf{F}_j^{\mathrm{drag}}$, respectively. The bond forces $\mathbf{F}_i^{\mathrm{bond}}$ and $\mathbf{F}_j^{\mathrm{bond}}$ represent the force contributions from element $j$ to $i$. Equal and opposite bond forces exist from element $i$ to $j$.

Following Herman (2013) and other prior sea ice DEMs, the ocean drag force on element $i$ is given by

$$\mathbf{F}_i^{\mathrm{drag}} = \rho^{\mathrm{o}} C^{\mathrm{hw}} A^{\mathrm{grain}} |\mathbf{U}_i - \mathbf{v}_i| (\mathbf{U}_i - \mathbf{v}_i), \tag{1}$$

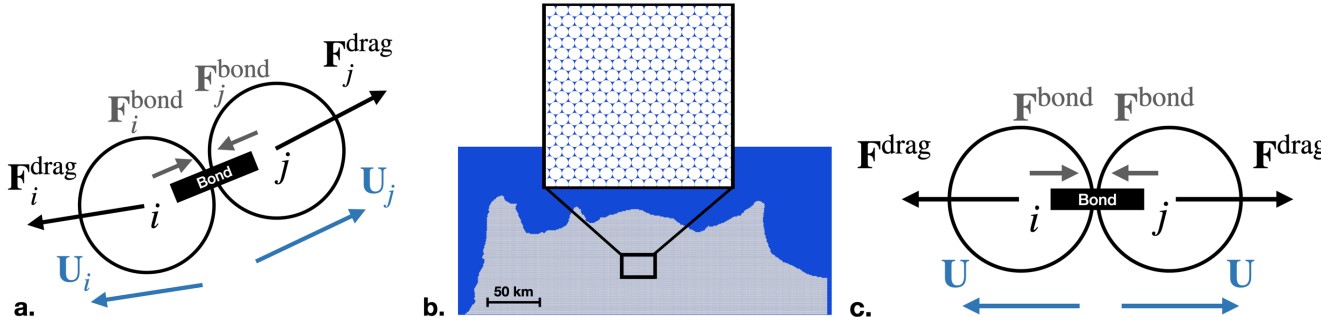

**Figure 2.** Application of BPM: a. A pair of sea ice grains joined by a bond. Relationship between the ocean currents $\mathbf{U}_i$ and $\mathbf{U}_j$ acting at each grain and respective drag forces at these elements. b. Example of an arbitrary bonded fast sea ice pack from Fram Strait with a hexagonal packing based on MODIS image on April 21, 2023 and the same coordinates in Fig. 1. c. Simplified ocean current conditions on a floe pair.





where $\mathbf{U}_i$ is the ocean velocity at the centroid of grain $i$, $\rho^{\mathrm{o}}$ is the ocean density, $C^{\mathrm{hw}}$ is the skin drag coefficient between sea ice and ocean water, and $A^{\mathrm{grain}}$ is the surface area of the discrete element. The model does not include form drag forces, as those are negligible for elements with a large length-to-height aspect ratio, as considered here.

To keep the analysis simple, and given the small ratio between thickness and area, we assume that bending moments do not exert significant bond forces. Then, the sheet is only subject to in-plane axial and shear stresses. As sea ice has very low critical

stresses for tension or shear, pure compression bond breakage or crushing failure is assumed to be unlikely. As a result, both bending and compressional failure are disabled in our simulations, and we focus on tension and shear failure modes.

Bond forces are generated when grain velocities $\mathbf{v}_i$ and $\mathbf{v}_j$ induce a relative velocity $\mathbf{v}^{\mathrm{rel}} = \mathbf{v}_i - \mathbf{v}_j$, which are specified in the normal or tension direction (parallel to bond) as $\mathbf{v}^{\mathrm{rel,n}}$ and in the shear direction (perpendicular to bond) as $\mathbf{v}^{\mathrm{rel,s}}$. The relative velocity induces a relative displacement $\mathbf{v}^{\mathrm{rel}}\Delta t$, which gives the bond force increment over a time step $\Delta t$:

$$\Delta\mathbf{F}^{\mathrm{bond,n}} = K^{\mathrm{bond,n}}\mathbf{v}^{\mathrm{rel,n}}\Delta t, \qquad \Delta\mathbf{F}^{\mathrm{bond,s}} = K^{\mathrm{bond,s}}\mathbf{v}^{\mathrm{rel,s}}\Delta t, \tag{2}$$

where $K^{\mathrm{bond,n}} = (EA^{\mathrm{bond}})/d^{\mathrm{coh}}$ and $K^{\mathrm{bond,s}} = (GA^{\mathrm{bond}})/d^{\mathrm{coh}}$ are the corresponding normal and shear bond stiffness, $E$ is the Young's modulus of sea ice, $G$ is the shear modulus, and $d^{\mathrm{coh}}$ is the maximum bonding or cohesive distance of the bond. Critical normal and shear forces on a bond are given in terms of a critical normal stress $\sigma^{\mathrm{c}}$ and a critical shear stress $\tau^{\mathrm{c}}$, respectively, and a cross-sectional bond area $A^{\mathrm{bond}} = \pi(r^{\mathrm{bond}})^2$, where bond radius $r^{\mathrm{bond}}$ is assumed to be equal to half the

sea ice thickness. This assumption allows us to relate bond critical strength to sea ice thickness. Specifically, we have

$$F_{\mathrm{n}}^{\mathrm{c}} = \sigma^{\mathrm{c}}A^{\mathrm{bond}}, \qquad F_{\mathrm{s}}^{\mathrm{c}} = \tau^{\mathrm{c}}A^{\mathrm{bond}}. \tag{3}$$

The bond is broken if at least one of these critical forces is exceeded, that is, if $|\mathbf{F}^{\mathrm{bond,n}}| > F_{\mathrm{n}}^{\mathrm{c}}$ or $|\mathbf{F}^{\mathrm{bond,s}}| > F_{\mathrm{s}}^{\mathrm{c}}$. For simplicity, we take the same value for the critical strength $F^{\mathrm{c}}$ under normal and shear forces. The time stepping of velocity based on the drag force includes a global damping term, as is customary in DEMs. Global damping is often used to account for dissipative energy processes in the system, such as inelastic collisions, solid-fluid momentum exchange, or inelastic deformations, all of

which remove energy from the system in ways not accounted for by particle or bond elastic stresses (Kawamoto et al., 2016).

## 3 Analytical solutions for breakage under uniaxial tension

In this section, we aim to establish a simplified analytical relationship between ocean current properties and bond breakage under uniaxial tension. These insights are later used to interpret the behavior of the sea ice pack under more realistic oceanic

forcing. We first derive an expression for the bond force and then obtain an estimate of the breakage length scale.

### 3.1 Uniaxial bond force

We consider the breakage of a single bond joining two floes that are initially stationary and an oceanic forcing that acts to stretch the bond in a direction parallel to the bond (Fig. 2c). We assume that prior to breakage, the ocean velocity is much





larger than the floe velocity ($|\mathbf{U}_i| \gg |\mathbf{v}_i|$), such that the drag force from Eq. (1) can be approximated as:

$$\mathbf{F}^{\mathrm{drag}} = \rho^{\mathrm{o}} C^{\mathrm{hw}} A^{\mathrm{grain}} \mathbf{U}_i |\mathbf{U}_i|. \tag{4}$$

The bond force is then purely uniaxial and can be written as

$$\mathbf{F}^{\mathrm{bond}} = K^{\mathrm{bond}} \Delta l, \tag{5}$$

where $\Delta l$ is the relative displacement between two floes and $K^{\mathrm{bond}} = K^{\mathrm{bond,n}}$. For a strong enough oceanic forcing, we may assume that $|\mathbf{F}^{\mathrm{drag}}| \gg |\mathbf{F}^{\mathrm{bond}}|$, such that the momentum balance for floe $i$ becomes

$$m_i \frac{du_i}{dt} = \mathbf{F}^{\mathrm{drag}}, \qquad \text{with} \quad m_i = \rho^{\mathrm{ice}} A^{\mathrm{grain}} h^{\mathrm{ice}}, \tag{6}$$

where $u_i$ is the velocity of the floe, $\rho^{\mathrm{ice}}$ is the density of sea ice, and $h^{\mathrm{ice}}$ is the thickness of the floe. We solve for $u_i$ by integrating over time, such that

$$u_i = \gamma |\mathbf{U}_i|(\mathbf{U}_i) t, \qquad \text{with} \quad \gamma = \frac{\rho^{\mathrm{o}} C^{\mathrm{hw}}}{\rho^{\mathrm{ice}} h^{\mathrm{ice}}}. \tag{7}$$

We now define the floe displacement relative to its initial position as $\epsilon_i$, with $u_i = d\epsilon_i/dt$. Integrating in time, we obtain

$$\epsilon_i = \int_0^t u_i \, d\tau = \int_0^t \gamma |\mathbf{U}_i|(\mathbf{U}_i) \tau \, d\tau = \frac{\gamma}{2} |\mathbf{U}_i|(\mathbf{U}_i) t^2, \tag{8}$$

The relative floe displacement may then be expressed as $\Delta l = \epsilon_j - \epsilon_i = \Delta \epsilon$. We now link the floe displacements expressed in a Lagrangian formulation to the forcing, which will be expressed in an Eulerian framework. We use a coordinate system consisting of the direction $x$ oriented parallel to the bond with the origin at the halfway point between the two floes. Assuming that the initial spacing between floes is equal to twice their radius $2r$, we may define an infinitesimal increment $\Delta x = 2r$ for $r \to 0$. In this limit, the bond force magnitude becomes

$$F^{\mathrm{bond}} = |\mathbf{F}^{\mathrm{bond}}| = r K^{\mathrm{bond}} \gamma t^2 \frac{d}{dx} \left( U_i^2 \right). \tag{9}$$

Eq. (9) shows that the bond force under uniaxial tension scales with the local gradient of the squared ocean velocity, and increases quadratically in time.

## 3.2 Breakage length scale under a pulse forcing

We now investigate the response of a sea ice pack subject to a uniaxial tensile pulse in 1D, as illustrated in Fig. 2c. We assume that the pack consists of a row of floes aligned in the $x$ direction and that each floe is bonded to its two neighbors. We also assume, as in Section 3.1, that the drag forces are much larger than the bond forces everywhere. Our goal is to derive an expression for the region over which the critical bond force is exceeded, and hence where we expect breakage to occur. For





mathematical convenience, we define the velocity pulse as half a sinusoidal wave using the piecewise scalar function

$$U_i(x) = \begin{cases} -U_{\mathrm{o}} & \text{if } x < -\lambda/4, \\ U_{\mathrm{o}} \sin \frac{2\pi x}{\lambda} & \text{if } -\lambda/4 \leq x \leq \lambda/4, \\ U_{\mathrm{o}} & \text{if } x > \lambda/4, \end{cases} \tag{10}$$

where $\lambda$ is the wavelength of the sinusoidal pulse, representing a characteristic forcing length scale and $U_{\mathrm{o}}$ is the velocity amplitude of the sinusoidal function.

Using Eq. (9), we evaluate the bond force for $-\lambda/4 \leq x \leq \lambda/4$, which becomes

$$F^{\mathrm{bond}}(x,t) = 2r K^{\mathrm{bond}} \gamma t^2 U_{\mathrm{o}}^2 \frac{\pi}{\lambda} \sin \frac{4\pi}{\lambda} x. \tag{11}$$

The location at which this bond force reaches the critical bond strength is thus

$$x^{\mathrm{c}} = \lambda \frac{1}{\pi} \arcsin\left( \frac{F^{\mathrm{c}} \lambda}{2\pi r K^{\mathrm{bond}} \gamma t^2 U_{\mathrm{o}}^2} \right), \quad \text{with} \quad U_{\mathrm{o}} \geq \sqrt{\frac{F^{\mathrm{c}} \lambda}{2\pi r K^{\mathrm{bond}} \gamma t^2}}. \tag{12}$$

As illustrated in Fig. 3a, the critical force is exceeded in two isolated zones, located between $x^{\mathrm{c}}$ and $\lambda/4 - x^{\mathrm{c}}$, and between $-\lambda/4 + x^{\mathrm{c}}$ and $-x^{\mathrm{c}}$, respectively. The existence of these two regions occurs because the critical bond force depends on the gradient of the squared velocity, rather than simply the velocity gradient.

The breakage length, defined as the extent in the $x$ direction over which the bond force exceeds the critical force, is then

$$l^{\mathrm{br}} = \frac{\lambda}{2} - 4x^{\mathrm{c}} = \lambda \left[ \frac{1}{2} - \frac{1}{\pi} \arcsin\left( \frac{F^{\mathrm{c}} \lambda}{2\pi r K^{\mathrm{bond}} \gamma t^2 U_{\mathrm{o}}^2} \right) \right]. \tag{13}$$

For large enough time, $l^{\mathrm{br}}$ in Eq. (13) tends to $\lambda/2$, as long as the forcing magnitude is strong enough ($r K^{\mathrm{bond}} \gamma t^2 U_{\mathrm{o}}^2 >> F^{\mathrm{c}} \lambda/2\pi$). In this case, $l^{\mathrm{br}}$ simply scales linearly with $\lambda$. However, breakage can occur over a smaller region than $\lambda/2$ if the forcing magnitude is weaker (but still strong enough to exceed $F^{\mathrm{c}}$). Using Eq. (13), we can investigate how $l^{\mathrm{br}}$ varies with the
forcing length scale and magnitude, as represented by $\lambda$ and $U_{\mathrm{o}}$, respectively (Fig. 3b). At a given time $t$, $l^{\mathrm{br}}$ first increases with $\lambda$ until it reaches a peak, after which $l^{\mathrm{br}}$ decreases until it reaches a cutoff value beyond which no breakage occurs. The initial increase in $l^{\mathrm{br}}$ is due to a larger area over which the forcing can act to break bonds. However, as $\lambda$ increases further, the ocean velocity gradient decreases, which reduces the bond force and leads to a smaller breakage area. Eventually, the forcing becomes too diffuse to exceed the critical bond force at any point, such that breakage is no longer possible. Using Eq. (13), we
find that this cutoff wavelength is

$$\lambda^{\mathrm{cut}} = \frac{2\pi r K^{\mathrm{bond}} \gamma t^2 U_{\mathrm{o}}^2}{F^{\mathrm{c}}}. \tag{14}$$

The wavelength that maximizes $l^{\mathrm{br}}$ can be found numerically by solving for the maximum breakage length $l^{\mathrm{br}}$ in Eq. (13). When increasing $U_{\mathrm{o}}$, the values of the peak and cutoff wavelengths tend to increase quadratically.





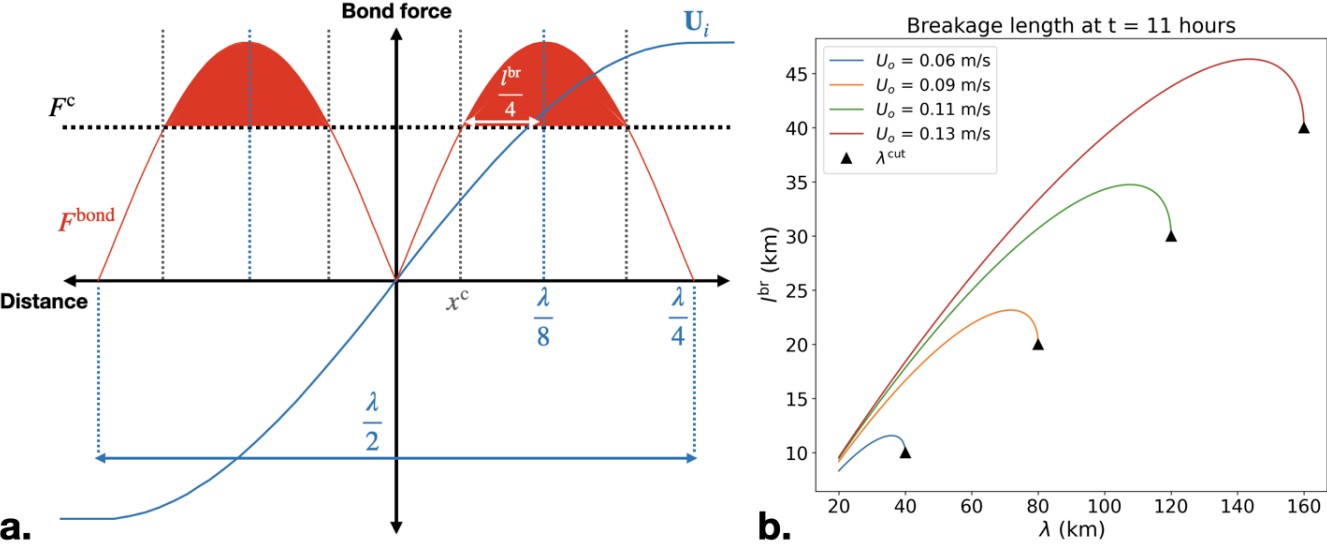

**Figure 3.** a. Diagram showing the ocean velocity and bond force under the uniaxial forcing defined via Eq. (10) and (9), respectively. The red shading indicates the two zones where $F^{\mathrm{bond}}$ exceeds $F^{\mathrm{c}}$. b. $l^{\mathrm{br}}$ as a function of $\lambda$ for different values of $U_{\mathrm{o}}$, as evaluated from Eq. (13) at $t = 11$ hours. The solid black triangles show the cutoff wavelength $\lambda^{\mathrm{cut}}$ from Eq. (14), beyond which no breakage occurs. The relevant parameters are provided in Table 1.

## 4 Uniaxial tension simulations

### 4.1 Setup

This section investigates the behavior of LS-DEM (Sec. 2) under the pulse forcing scenario described in Sec. 3. The sea ice pack geometry consists of a square 100 km × 100 km sheet of floes placed at the center of the domain in the $x$-direction, such that half of the floes are pulled eastward and the other half are pulled westward. The sea ice pack is not fixed at any point. The ocean velocity is constant in time and given by Eq. (10). For these idealized simulations, the damping term in LS-DEM is turned off, allowing a more direct comparison with the analytical solutions from Sec. 3. We use values of sea ice stiffness and drag coefficient that are similar to previous studies (Herman, 2016; Timco and Weeks, 2010), as summarized in Table 1. The critical bond strength is adjusted slightly to allow gradual breakage within the simulated time, which is 11 hours.

We conduct a sensitivity analysis by varying the wavelength of the forcing $\lambda$ and the ocean velocity magnitude $U_{\mathrm{o}}$. For each simulation, we estimate the simulated breakage length $l^{\mathrm{br}}_{\mathrm{sim}}$ by evaluating the distance between the furthest two broken bonds for each row in the $x$ direction and then averaging over all rows. These breakage lengths are computed using the original positions of the broken bonds at $t = 0$, to avoid the subsequent floe motions to bias the result.



**Table 1.** Parameters for the numerical simulations forced by uniaxial pulses.

| Variable | Meaning | Value | Units |
|:---:|:---:|:---:|:---:|
| $K^{\mathrm{bond}}$ | Bond Stiffness | $6 \times 10^9$ | $\mathrm{N\ m^{-1}}$ |
| $\sigma^{\mathrm{c}}$ | Bond critical normal strength | $1.5 \times 10^4$ | Pa |
| $\Delta t$ | Time step size | 0.92 | s |
| $r$ | Grain radius | 0.5 | km |
| $d^{\mathrm{coh}}$ | Cohesive distance | 0.1 | km |
| $h^{\mathrm{ice}}$ | Sea ice thickness | 1.0 | m |
| $C^{\mathrm{hw}}$ | Water-ice skin drag coefficient | $10^{-3}$ | - |

## 4.2 Results

We first present results pertaining to two simulations representing a relatively sharp ($\lambda = 25$ km) and diffuse ocean forcing ($\lambda = 100$ km), respectively, for the same ocean velocity magnitude ($U_{\mathrm{o}} = 0.03$ m s$^{-1}$). When $\lambda = 25$ km, the breakage zone is narrow ($l_{\mathrm{sim}}^{\mathrm{br}} \sim 12.5$ km), and most bonds are broken within that region (Fig. 4a). Most of the bonds that remain unbroken are found around the origin ($x = 0$), as the bond force is relatively weak there (Fig. 3a). When $\lambda = 100$ km, the breakage zone is wider ($l_{\mathrm{sim}}^{\mathrm{br}} \sim 100$ km), but a smaller fraction of bonds are broken within that region than for the sharper pulse (Fig 4 b). There are two clear breakage lines around $x = \pm \lambda / 8$, which is where the bond force is the highest (Fig. 3a). In both of these cases, $l_{\mathrm{sim}}^{\mathrm{br}}$ is approximately $\lambda / 2$, which corresponds to the largest possible breakage length for the given forcing (Eq. 13).

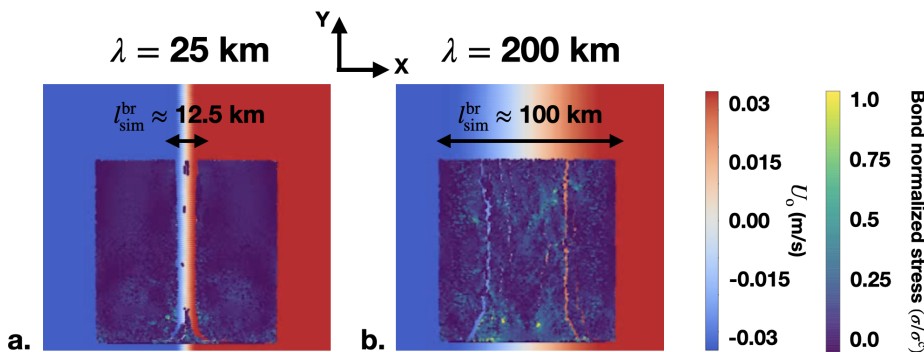

**Figure 4.** Breakage of a square sheet of sea ice under the uniaxial tension forcing of Eq. (10) with $U_{\mathrm{o}} = 0.03$ m s$^{-1}$ and after 11 hours, for a. $\lambda = 25$ km and b. $\lambda = 200$ km. The background colors show the prescribed ocean velocity. Individual bonds from the DEM are colored by their stress $\sigma$, normalized by the critical stress $\sigma^{\mathrm{c}}$. The simulation parameters are summarized in Table 1.

To test the validity of the formulations derived in Section 3, we compare the analytical ($l^{\mathrm{br}}$) and simulated ($l_{\mathrm{sim}}^{\mathrm{br}}$) breakage lengths for simulations where we vary $U_{\mathrm{o}}$ between 0.1 and 1 m s$^{-1}$ in increments of 0.1 m s$^{-1}$ (Fig. 5). We find that $l_{\mathrm{sim}}^{\mathrm{br}}$ is consistently lower than $l^{\mathrm{br}}$, though the difference between the two quantities decreases for increasing $U_{\mathrm{o}}$, such that both tend to





$\lambda/2$ for large forcing magnitude. The overestimate of the breakage length by the analytical solution may be due to the fact that it does not account for the bonds connecting individual $x$-direction rows, which carry both tensile and shear stresses. Moreover, as the floes move in response to the forcing and the first bonds start to break, the bond forces are redistributed within the pack, forming complex stress patterns that include points of highly localized stress concentration (Fig. 4). Boundary effects at the edge of the pack may also play an important role.

The analytical formulations discussed in Section 3 help us obtain a qualitative understanding of the breakage behavior under uniaxial tension. When the forcing is more strongly concentrated (short wavelength), the breakage region is small and forms distinct fracture lines. On the other hand, when the forcing is more diffuse (large wavelength), the breakage is spread over a larger area, but the fracture lines are less well distinguishable. The analytical $l^{\mathrm{br}}$ provides an upper limit to $l^{\mathrm{br}}_{\mathrm{sim}}$, but ignores the more complex floe network dynamics that play a significant role in the DEM behavior, even under a relatively simple forcing scenario. In the real ocean-ice system, the forcing is highly heterogeneous, such that non-linear dynamics are likely even more significant than for uniaxial tension. We explore the DEM's behavior for such ocean currents in the following section.

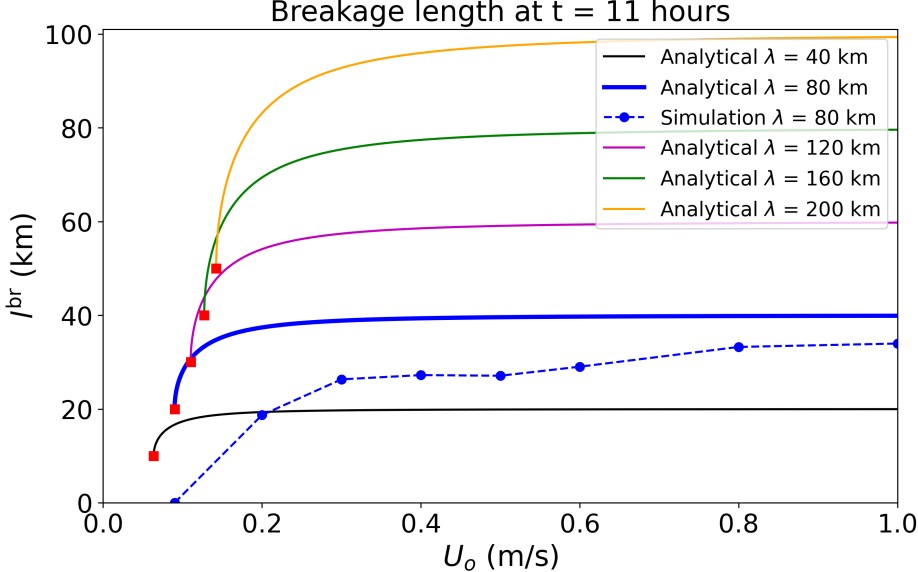

**Figure 5.** Effect of ocean velocity amplitude on breakage length with the values of Table 1 at 11 hours after the application of the fixed pulse. Comparison of simulation results (dotted line) and analytical expression Eq. (13) (continuous lines) for a fixed wavelength $\lambda = 80$ km. The limit velocity for each analytical curve $U^{\mathrm{limit}}$ is represented by the red solid squares obtained by solving for velocity in Eq. (14).





## 5 Eddying currents simulations

### 5.1 Setup

We now consider the response of the pack to more realistic ocean currents, consisting of eddies generated by a two-layer quasigeostrophic (QG) model (Haidvogel and Held, 1980; Bluestein, 2006a, b; Medjo, 2008), as implemented in the `pyqg` python package (Jansen et al., 2015). Ocean velocity snapshots derived from the top layer of the QG model are stored every 6 hours and are used to force the LS-DEM model, with no feedback from the ice to the QG model. The properties of the currents generated by the QG model are further discussed in Section 5.2. Simulation parameters pertaining to the DEM are summarized in Table 2. In the literature, critical sea ice strength ($\sigma^{\mathrm{c}}$) can range between 100 kPa and 20 MPa, depending on the brine content and failure mode (Herman, 2016; Timco and Weeks, 2010). Here, we use values between 7.5 to 15 MPa for normal strength, which are on the higher end of this range. These large values allow sufficient time to observe breakage within the simulated time (48 days) and keep the computational expense reasonable for our sensitivity analysis.

The geometry of the simulations consists of a square domain with sea ice present in its southern part. The edge of the sea ice cover is irregularly shaped, based on a visible image snapshot from the NASA MODIS satellite sensor taken in Fram Strait on May 28, 2023. An outline of the sea ice pack is meshed using the hexagonal packing of the DEM, as illustrated in Fig. 2b. The southern-most row of sea ice grains is constrained to zero displacements, which emulates the behavior of fast ice anchored to the shore. The purpose is not to represent the comprehensive behavior of the pack in that region, but to obtain a realistic edge geometry, which facilitates the formation of cracks under the eddy forcing.

Underneath sea ice, surface ocean velocities are typically damped due to ice-ocean friction (Gupta et al., 2020; Manucharyan and Thompson, 2022). Our one-way coupled setup does not explicitly capture this effect, since the QG model does not feel any friction that would be induced by the sea ice pack. We emulate this effect using a simple formulation, where the ocean velocity is damped exponentially away from the sea ice edge, as follows:

$$\mathbf{U}_i^{\mathrm{damp}} = \begin{cases} \mathbf{U}_i \exp\left(\frac{y^{\mathrm{edge}} - y_i}{D^{\mathrm{s}}}\right) & \text{if } y_i < y^{\mathrm{edge}}, \\ \mathbf{U}_i & \text{if } y_i \geq y^{\mathrm{edge}}, \end{cases} \tag{15}$$

**Table 2.** Parameters for the numerical simulations with eddying currents from the QG model (omitted parameters are the same as in Table 1).

| Variable | Meaning | Value | Units |
|---|---|---|---|
| $K^{\mathrm{bond}}$ | Bond Stiffness | 30e9 | N/m |
| $\sigma^{\mathrm{c}}$ | Bond critical normal strength | 7.5e6, 10.5e6, 15e6 | Pa |
| $\delta t$ | Time step size | 10 | s |
| $D^{\mathrm{s}}$ | Ice damping distance | 2.5 | km |
| $h^{\mathrm{ice}}$ | Sea ice thickness | 0.7, 1.0, 1.4 | m |



where $\mathbf{U}_i$ is the original QG velocity at a grid cell $i$, $\mathbf{U}_i^{\mathrm{damp}}$ is the corresponding damped velocity, $y_i$ is the $y$-center of that grid cell $i$, $y^{\mathrm{edge}}$ is the mean sea ice edge, and $D^{\mathrm{s}}$ is a characteristic damping distance set to 2.5 km. Ocean cells that do not contain sea ice or whose overlaying ice floes are not connected to the landfast ice are not affected by the damping scheme.

At any instant in time, $y^{\mathrm{edge}}$ is defined as the $y$-centroid of the landfast sea ice pack, increased by 30 km to account for the distance between the centroid and the sea ice edge. The conclusions of our work are not too sensitive to this definition, since results without this kind of damping follow the same trends.

## 5.2 Eddying current properties

The simulations from the QG model are first run until a quasi-equilibrium is reached, before the resulting eddying currents are
used as a forcing to the DEM. These numerical experiments are conducted in a doubly-periodic domain of dimensions equal to the DEM simulations, namely 400 km $\times$ 400 km, with a resolution of 1.56 km. The ratio between the top and bottom layer thickness of the QG model is $\delta = 0.25$ and the gradient of the Coriolis frequency is $\beta = 1.5 \cdot 10^{-11}$ m$^{-1}$s$^{-1}$. We consider two different scenarios, where the radius of deformation $r^{\mathrm{d}}$ is set to 5 and 30 km, representing a small ($r_5^{\mathrm{d}}$) and a large ($r_{30}^{\mathrm{d}}$) eddy case, respectively. For a better comparison between the two scenarios, the upper layer flow background mean velocity $U_1$
is adjusted until the two simulations display comparable eddy kinetic energy (EKE), over the duration of the simulation and giving 0.05 and 0.015 ms$^{-1}$ for $r_5^{\mathrm{d}}$ and $r_{30}^{\mathrm{d}}$, respectively (Fig. 6a). The lower-layer background mean velocity is set to zero in both cases.

In the quasi-equilibrated state, the power spectrum of the EKE displays more energy at shorter wavelengths for $r_5^{\mathrm{d}}$, while there is more energy at larger wavelengths for $r_{30}^{\mathrm{d}}$ (Fig. 6b). In the inertial range, the slope of the spectrum is approximately
-3.5 for both $r_5^{\mathrm{d}}$ and $r_{30}^{\mathrm{d}}$. We can define a weighted-average wavelength from the EKE spectrum as follows:

$$\lambda^{\mathrm{ave}} = \left( \int_{k_{\min}}^{k_{\max}} \frac{E(k)k}{E(k)} dk \right)^{-1}, \tag{16}$$

where $E(k)$ is the EKE density at wavenumber $k$. We find that the time-average value of $\lambda^{\mathrm{ave}}$ is 7.49 km and 27.25 km for $r_5^{\mathrm{d}}$ and $r_{30}^{\mathrm{d}}$, respectively, which matches relatively well with their prescribed deformation radii values. Based on the analytical model explored in Section 3, we thus expect breakage over narrower regions for $r_5^{\mathrm{d}}$ and a more extended breakage zone for
$r_{30}^{\mathrm{d}}$.



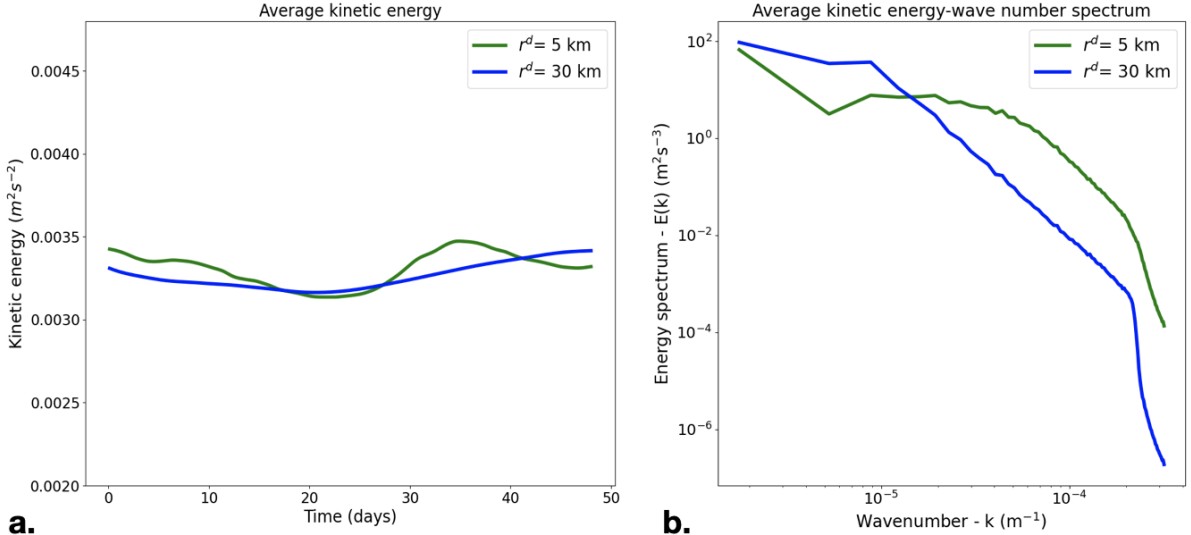

**Figure 6.** Properties of the QG model currents for $r^{\mathrm{d}} = 30$ km (blue) and $r^{\mathrm{d}} = 5$ km (green). a. Evolution of the EKE averaged over the domain. The EKE was designed to be similar between the two simulations by varying the forcing ocean velocity. b. Power spectrum density of the EKE averaged over all time steps.

## 5.3 Results

Snapshots of the ocean currents and sea ice pack help us visualize and compare the fracturing process for $r^{\mathrm{d}} = 5$ km and $r^{\mathrm{d}} = 30$ km. (Fig. 7). Near the start of the simulations, the breakage mostly occurs around sharp asperities of the sea ice edge, which is where we expect the highest stress concentration to develop. At subsequent time steps, the fracturing spreads deeper

into the pack, pushing the mean sea ice edge further southward. The penetration of breakage is due to a combination of internal stress waves that reach the interior of the pack, as evidenced by cracks in the fast ice area, and the reduced damping of ocean currents under ice as sea ice breaks. Visual inspection suggests that breakage penetrates deeper in the pack for $r^{\mathrm{d}}_{30}$ as compared to $r^{\mathrm{d}}_{5}$, consistent with a larger breakage area for forcings of larger wavelength (Section 3). Fig. 7 also shows individual floes, where a floe is defined as a collection of sea ice grains that are linked via bonds. From the start of the simulations, ocean eddies

generate a variety of floe shapes and sizes. In some regions, particularly towards the end of the simulations, some floes become composed of a single grain, such that breakage can occur even at the finest allowable scale of the DEM.



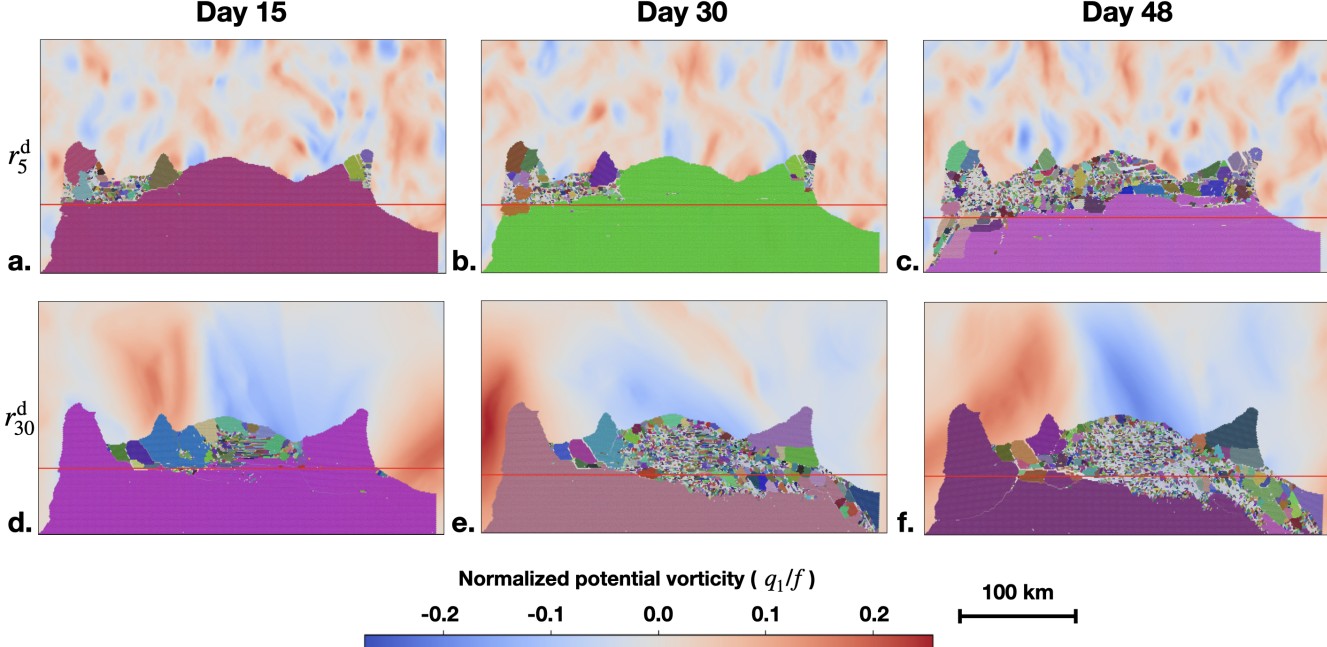

**Figure 7.** (a–f) Snapshots of the upper layer QG potential vorticity normalized by the Coriolis frequency ($q_1/f$, background colors) super-imposed with the sea ice pack represented by the DEM. Individual sea ice floes, as detected by the `igraph` library in Python (Csárdi et al., 2024), are colored in different shades, where white represents floes composed of only one DEM element. The simulations plotted here have $\sigma^c = 10.5$ MPa. We consider snapshots at day 15, 30, and 48 of the simulation for $r_5^d$ (a–c) and $r_{30}^d$ (d–f). The red horizontal line shows the location $y^{edge}$ used in Eq. (15) to emulate the damping of ocean currents.

We next evaluate a variety of metrics to quantify the breakage characteristics of the pack, and compare simulations with different forcing length scales ($r^d = 5$ and 30 km) and sea ice bond strength ($\sigma^c = 7.5$, 10.5 and 15 MPa). We first consider the time-evolution of the fast-ice area, namely the area of ice that is still connected to the stationary floes in the south of the domain

(Fig. 8a). Simulations with $r^d = 30$ km evolve into a pack that has less consolidated sea ice than for $r^d = 5$ km at all values of $\sigma^c$. The reduction in fast-ice area tends to occur in bursts, with relatively inactive periods representing the formation of cracks, followed by a sharp decline representing the detachment of individual floes from the main pack. As expected, a greater bond strength leads to less breakage and a larger fast-ice area.

The number of floes and their size distribution are also useful metrics for characterizing the breakage process. The total

number of floes increases from 1 to several thousand over the course of the simulations (Fig. 8b). This increase is larger for simulations with $r^d = 30$ km compared to $r^d = 5$ km, at all values of $\sigma^c$, reflecting a larger number of floes being detached from the fast-ice and further broken down into smaller pieces by the eddying currents. As with the total fast-ice area, greater values of $\sigma^c$ lead to a smaller number of floes generated in the domain. Despite the differences in the number of floes produced by breakage, the FSD slope is only slightly (but appreciably) steeper for $r^d = 30$ km, as compared to $r^d = 5$ km. Larger

differences in the FSD slope are seen when varying the bond strength, with steeper slopes recorded for smaller values of $\sigma^c$.





Nevertheless, the breakage mechanism differs based on the radius of deformation. For $r^{\mathrm{d}} = 30$ km, a greater fraction of the bonds tend to break due to tension compared to shear, while the opposite is true for $r^{\mathrm{d}} = 5$ km. This difference holds for all values of $\sigma^{\mathrm{c}}$.




**Figure 8.** Sensitivity of various metrics to the ocean current's radius of deformation, which is also associated with the eddy scale (green: $r^{\mathrm{d}} = 5$ km & blue: $r^{\mathrm{d}} = 30$ km) and the bond strength $\sigma^{\mathrm{c}}$ (stars: 15 MPa, dashed: 10.5 MPa, solid: 7.5 MPa). (a) Fast-ice area. (b) Number of floes. (c) Cumulative FSD averaged over the last 20 days of the simulations. Note that this FSD does not include the landfast portion of the pack. The FSD slope $\alpha$ calculated using the Maximum Likelihood Estimate method of Virkar and Clauset (2014) is indicated in the legend and is expressed in absolute value. (d) Ratio between the number of bonds that break due to shear versus tension, as defined by which threshold is exceeded first. The horizontal dashed line indicates a ratio of 1 or same proportion of normal and shear breakage.



## 6 Discussion

This study has investigated the process of sea ice breakage due to ocean eddies, first using a simple analytical model and then
a DEM forced by prescribed, turbulent ocean currents. The analytical formulation predicts that the area of sea ice affected
by breakage depends on the gradient of the squared ocean velocity and that there is an optimum forcing length scale that
maximizes the breakage area for a given velocity magnitude. The DEM shows qualitatively similar behavior to the analytical
model under uniaxial tensile forcing, but also more complex effects, such as 2-D stress redistribution after fracture, that likely
explains the quantitative differences between the model and theory.

When forced by ocean eddies generated by a QG model, the DEM produces floe clusters and fracture lines that are reminis-
cent of those visible in satellite images of the marginal ice zone (MODIS Science Team, 2017). The breakage occurs across all
modeled scales, including the smallest allowable one by the DEM, namely the imposed grain radius (1 km). The emergent FSD
slope is within the observed range (Stern et al., 2018; Buckley et al., 2024), and does not vary strongly between the $r^{\mathrm{d}} = 5$ km
and 30 km simulations. Consistent with the analytical results, the eddying currents penetrate deeper in the pack for $r^{\mathrm{d}} = 30$
km, leading to a greater reduction in the fast ice area. We note that these two forcings have a similar turbulent inertial range,
while in reality the influence of sub-mesoscale processes can alter this slope (Manucharyan and Thompson, 2017; Gupta and
Thompson, 2022), and may therefore also affect the breakage mechanism and the resulting FSD.

The discrete element method employed in this work is distinct from the continuum formulation typically used for modeling
sea ice at kilometer scales and above. These continuum formulations, when run at sufficiently high resolution, have shown the
ability to form realistic crack patterns, notably in the interior sea ice pack (Bouchat et al., 2022; Hutter et al., 2022). However,
the granular behavior of sea ice in marginal ice zones is more difficult to simulate and validate with the continuum approach.
As shown in this work and others (Moncada et al., 2023; Åström et al., 2024; Moncada et al., 2025), DEMs can form sharp
crack lines and distinct floes that can be readily compared with high-resolution observations.

The simulations presented here were idealized and do not represent some processes that are important for the breakage in the
marginal ice zone. For time scales relevant to mesoscale eddy processes (days to weeks), sea ice thermodynamics, including
changes in the snow cover, play a critical role in controlling the strength and distribution of sea ice (Sledd et al., 2024).
Moreover, the finite minimum floe size considered here (1 km) does not allow us to simulate finer scale leads and can affect
stress distribution at the larger scale. Selecting the appropriate DEM parameters is also a challenge, as the model behavior can
be sensitive to these choices. The lack of coupling between the ocean currents and the sea ice pack also affects our results, as
we expect individual sea ice floes to have important local effects on ocean currents (Gupta and Thompson, 2022; Gupta et al.,
2024), which can feed back on the breakage process.

The mesoscale eddies considered in this work represent only a subset of the various forcings relevant to sea ice breakage
in the marginal ice zone. Ocean waves are a primary driver of this fracturing process via vertical motions (Squire et al., 1995;
Langhorne et al., 1998; Yang et al., 2024), but were not considered in this work, as it focused on horizontal motions due to
eddies. Large-scale ocean currents can also play a significant role in fracturing the sea ice pack and in generating long-lasting
leads that can span several thousands of kilometers. At those scales, wind forcing, notably during powerful storms, have been



shown to considerably affect the concentration and thickness distribution of the pack with coupling to wave and ocean current behavior (Wang et al., 2021). Improving the DEM formulation to include these effects will be considered in future work.

## 7 Conclusions

This work presents the first simulations of sea ice breakage due to ocean eddies using a bonded discrete element model. In a configuration representing an idealized marginal ice zone with landfast ice, ocean eddies fracture the sea ice into individual floes with sizes ranging between a kilometer (the DEM grain size) and tens of kilometers. Simulations forced by currents that have a larger radius of deformation $r^{\rm d}$ (but the same EKE) break a larger fraction of the pack as compared to experiments

with a smaller $r^{\rm d}$. This behavior is qualitatively consistent with a simple analytical model, which shows that ocean currents with a larger characteristic wavelength tend to break sea ice over a larger area. The analytical breakage threshold depends on the gradient of the squared velocity, due to the quadratic drag relationship between ocean and sea ice. Tension-driven fracture dominates for a large $r^{\rm d}$ (30 km), while shear-driven breakage is most important for a smaller $r^{\rm d}$ (5 km). Despite the different fracturing mechanisms, the emergent FSD slope does not vary much with the forcing length scale, but instead depends most

strongly on the imposed bond strength.

In the coming decades, the Arctic is set to have an MIZ that covers a larger fraction of the total sea ice area (Frew et al., 2025), as well as a more energetic ocean (Muilwijk et al., 2024). Current climate models do not capture all the relevant mechanisms occurring in those regions, including the breakage of sea ice by ocean eddies. As these processes become more prominent, it is increasingly important to represent them accurately for global climate modeling and polar-specific forecasts. The insights

derived from this work may help better parametrize these effects, notably by incorporating them in existing formulations of the joint floe size and thickness distribution evolution, which currently focus largely on waves fracturing sea ice (Horvat and Tziperman, 2015; Montiel and Squire, 2017). The development of DEMs, such as the one used in this work, may also help with hybrid approaches that combine continuum models at the large-scale with discrete elements at finer scales (Blockley et al., 2020; Åström et al., 2024; Tsarau et al., 2024).

*Code and data availability.* MODIS images can be accessed from MODIS Science Team (2017). LS-DEM software for simulations is preserved and developed at GitHub: https://github.com/rmoncadalopez/LSICE.git.

*Author contributions.* 1. DEM and mechanical analysis, sea ice simulations, result processing. 2. Ocean currents and sea ice response, QG simulations. 3. Further mechanical analysis and exploration of breakage, 4. Advice on ocean currents and interpretation of results 5. Advice on DEM and granular mechanics.

*Competing interests.* All the authors declare having no conflict of interests with respect to the results of this paper.



*Acknowledgements.* R.M.L., J.U. and J.A.'s research was funded by the support of ARO Grant W911NF-19-1-0245 and the NSF grant JEA.NSFCMMIECI-1-NSF.2033779. M.G. and A.F.T. were supported by award NSF-OCE 1829969 and the Office of Naval Research Multidisciplinary University Research Initiative (MURI) on Mathematics and Data Science for Physical Modeling and Prediction of Sea Ice, N00014-19-1-2421. Portions of the manuscript were written under the auspices of the U.S. Department of Energy by Lawrence Livermore National Laboratory under Contract DE-AC52-07NA27344.




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
