# Peer review of "Simulating landfast sea ice breakage due to ocean eddies using a discrete element model"

_EGUsphere, 2025_

## Referee Comment (RC1)

**General comments**

The authors present an analytical derivation of the bond forces experienced by a one-dimensional assembly of particles under a velocity pulse located in the middle of the assembly. They show that it is expected that such an assembly would break in 1–2 points based on the slope of the velocity pulse. Next, the authors forced a DEM initialized based on MODIS imagery with the upper velocity layer of a quasi-geostrophic (QG) model. They performed a sensitivity analysis on different parameters of the DEM, as well as different deformation radii (using the same EKE) for the QG model. They demonstrated that the ice pack area decreases in size as the deformation radius increases. They also showed that the number of floes strongly depends on the bond strength, but that this does not reflect in the FSD, as all simulated FSDs appear somewhat similar. They showed that the difference was primarily due to a difference in failure mode (more tension failure for large deformation radius). The research conducted is meaningful and will likely be appreciated by the community. This manuscript contains good ideas, and we would like the authors to have the opportunity to publish it. **But, in this present condition, we cannot recommend this manuscript for publication. Instead, major revisions are recommended.**

One major drawback of the simulation tool used is the lack of compressive failure of sea ice. It is misleading to forget the various failure modes of sea ice and to refer only to crushing when justifying the exclusion of compressive failure. It is well established that pressure and shear ridges are prevalent wherever ice deformation occurs, indicating that sea ice indeed often fails under compressive loads. DEM tools capable of representing ridging have been used for more than two decades (see work by Hopkins); omitting compressive failure is not acceptable. Further concerns arise related to the parameterization of the simulations. The tensile strength of 15 MPa assigned to sea ice in the model is about 75 times higher than values typically measured in small-scale experiments, whereas large-scale values would be expected to be even lower. Similarly, the stiffness value of $30\,\mathrm{Nm^{-1}}$ is unrealistically high. The authors must reassess their assumptions, revise their parameterization, and likely rerun their simulations to ensure that the results are physically meaningful. For example, on lines 98–101, the authors disable the bending and compression bond-breaking schemes, arguing that these breaking modes are generally not reached.

We are now doubtful that the results presented in Section 5 are robust. There are very concerning questions about exactly how the model behaves that are not addressed in the text, notably, the fact that the bond stiffness is $K = 30 \times 10^9\,\mathrm{Nm^{-1}}$, which is a whole order of magnitude above the values typically used. In and of itself, this is not necessarily a problem, but considering the very high time step size ($10^1$ s) when compared to usual time step sizes in other DEMs ($\Delta t \leq 10^{-3}$ s), it raises the question whether the model is fully converged or not. Such a large combination of time step size and stiffness will inevitably lead to large overlaps, therefore, to large forces. We are surprised that the model does not explode at the edge. This means that the damping term must be very large, and it should be discussed in the text.

Furthermore, considering that the entire study focuses on the interactions between water and ice, it is surprising that the total drag forces were not included in the analysis. The authors specifically say on lines 96–97 that "The model does not include form drag forces, as those are negligible for elements with a large length-to-height aspect ratio, as considered here.", which may be an unjustified simplification. Body drag accounts for $\sim$ 14% of the total water drag on disk-shaped particles of 1 m thickness with 500 m radius. The ratio of form to surface drag in that case is morally $F_d/F_s = 2hC_d/(rC_s)$, and $C_s$ is around 2 orders of magnitude smaller than $C_d$, which brings this ratio to order 1 for particle radius of order $10^2$. Therefore, the authors have reduced stresses that propagate in the interior of their assembly. Indeed, these forces would only apply at the boundaries, but the stress still propagates, and it is not negligible. We don't see a valid reason as to why these forces would be overlooked. It is clearly not for computational efficiency, and including them would only strengthen the authors' argument and streamline the simulations.

Along the same lines, as one of the main results, the authors state that "for the same amount of eddy kinetic energy, ocean currents with a larger characteristic eddy size penetrate deeper into the pack and fracture more floes." However, the model used by the authors does not allow for such a conclusion. The model lacks two-way coupling between the ice and the currents and includes an ad hoc damping model for the current, where the user defines the damping distance. Very little can be said about the physics or the effect of eddy attenuation under the ice cover based on the model used. We recommend moving this conclusion and the

description of this numerical experiment from the paper.

On lines 179–180, the authors are turning off global damping: "For these idealized simulations, the damping term in LS-DEM is turned off, allowing a more direct comparison with the analytical solutions from Sec. 3." What type of damping are the authors referring to here? Depending on its type, damping should not be turned off in DEMs, especially with such high stiffness, time step size, and for 11 hours. What integration scheme are the authors using that justifies turning the damping off? Is the contribution from the damping accounted for in determining the stress state of the bond to decide if the bond fails, and is this the reason it has to be set off? It would be helpful to see the bond stresses over time and ensure that the breaking is not simply due to spurious oscillations of the forces.

Section 3 on breakage length under uniaxial tension is well-constructed and could prove to be an effective method for testing bond schemes in discrete element models. On a casual note, it is also the reason why uncooked spaghetti breaks in more than 2 parts most of the time when you bend it (Feynman's spaghetti problem). A similar analysis of the bending failure mode would reveal a similar behaviour.

However, special attention should be paid to the coherence between the math, the graphs, and the text. The results should be adapted to reflect these changes. The method presented in section 3.2, while interesting, is plagued with issues in the coherence of the math, graph, and text, which lead us to doubt the subsequent results. Indeed, Eqn. 13 does not lead to the graph presented in Fig. 3b (see specific comments below). The function should be zero when evaluated at $\lambda_{cut}$ as described in the text (eqn. 14).

Maybe a better approach, as highlighted by the authors in section 4.2, is to simply use the maximum of the bond force as a reference point (at $\lambda/8$) to benchmark whether a model's bond schemes work properly in tension. Moreover, based on the authors' own math/figure, you can't use $\lambda > 40\,\mathrm{km}$ for velocities $U_0 = 0.06\,\mathrm{m/s}$, hence how is a $\lambda = 200\,\mathrm{km}$ used for $U_0 = 0.03\,\mathrm{m/s}$? Shouldn't that be above the maximal allowed value as per equation 13? Section 4.2 will need to be rewritten to accommodate the changes required in Eqn 13. Furthermore, if the authors were concerned about reproducing the exact theoretical calculation with the model, they should have performed their validation experiments in Section 4 using a simple 1D rod of identical particles. They point out this exact issue on lines 198–199.

In equation 15, $y_{edge}$ is a free parameter, and the results (fig. 7) clearly are critically dependent on its value. The authors say, "The conclusions of our work are not too sensitive to this definition, since results without this kind of damping follow the same trends." Therefore, why include it at all?

In Section 5.3 and Figure 7, we have concerns about the simulation results. We can see these long, thin rods of particles stacked on top of themselves. This seems to point toward the fact that their bonds can only break in tension and shear. The fact that the "center" of the broken parts is entirely broken should be addressed in the text. Does that happen all at once, or is it building up? When examining Fig. 8, we can observe these sharp steps in the different panels. The authors note that the FSDs are all somewhat similar, despite the number of floes varying significantly. This is not something we would have expected, given the characteristics of the vorticity field. We would have assumed fewer larger floes and more small floes for $r_5^d$ when compared to the simulations with $r_{30}^d$. How is the radius computed in the FSD? Because all the resulting FSDs appear the same, this suggests that the model may simply explode in bursts rather than fracture. The fact that the mode of failure differs (tension vs. shear) does not give us confidence that the model is not exploding; it just does it differently.

**Specific comments**

**line 7:** "...a slightly higher FSD slope *when* compared ..." what FSD slope refers to isn't clear, please reformulate.

**line 9:** "...notably shallower FSD." what does shallower mean for a distribution?

**line 19:** "...by approximately one half ..." what are the units? Meters?

**lines 19–21:** This sentence does not flow very well, please reformulate.

**line 34:** "…arrested …", not sure arrested is the right choice of word.

**lines 34:** "…ice–ocean …" usage of an en dash in this case, please check throughout the manuscript.

**line 37:** "…wave–ice …" en dash.

**lines 40–42:** This sentence is hard to follow, please reformulate.

**line 85:** "A given pair of elements is connected via one bond at most, for simplicity." Isn't that a given? Because $F_{ij} = -F_{ji}$.

**eqn. 13:** This is a major comment. See above. We think we understand the authors' definition of the breakage length. Our understanding is that it is the length over which the bonds linking the particles in the row will be broken, effectively breaking the row into two or three parts. The length of the broken sections between the parts is given by that equation. Please clarify the text. Moreover, there is a factor of 4 missing in front of the arcsin, based on eqn. 12.

**fig. 3b:** This is a major comment. See above. The graph does not make sense with eqn. 13. Indeed, at $\lambda_{cut}$, the breakage length should be zero: when putting eqn. 14 in eqn. 13, one gets $l = \lambda(1/2 - \pi^{-1} \arcsin(1)) = 0$. Even considering the missing factor of 4, one still obtains negative values. Whatever is plotted here is not Eqn. 13, as presented. We suspect the following: the factor of 1/2 in eqn. 13 became 3/4. What was indicative of that is the fact that $\lambda_{cut} \approx \lambda/4$ in the figure, and this is the only way to achieve this. The fact that the equation is zero on the right side of the domain should be addressed in the text.

**line 163:** "$l^{br}$ simply scales linearly with $\lambda$, as long as the forcing magnitude is strong enough". This statement is true irrespective of the forcing applied (as long as you respect eqn. 12). If $t \to \infty$, then $l^{br} \to \lambda/2$. The $\gg$ sign inside the parentheses is confusing, because it seems to imply the following: when $rK\gamma t^2 U^2 \gg F^c \lambda/2 \implies \arcsin(x) = x + \mathcal{O}(x^3)$ around $x = 0$. And $l^{br} = \lambda/2 - A\lambda^2/\pi + \mathcal{O}(\lambda^4)$, where $A = F^c/(2rK\gamma t^2 U^2)$, which is not what the authors are saying in the text.

**lines 184–185:** We have a hard time understanding how you estimate the breakage length.

**line 191:** "$\lambda = 100\,\text{km}$" but in fig. 4b it says 200 km. Which is it?

**lines 193–194:** You would expect that to be the case, no? Because $F_{bond}$ is also a function of time, since it is based on the elongation, and the elongation needs a few time steps to build up. Once the stress is relieved, you would not expect new breakage lines to form. See above.

**lines 196–198:** "We find that $l^{br}_{sim}$ is consistently lower than $l^{br}$". That should come as no surprise, considering that Eqn. 13 is not what they used.

**lines 201–202:** "Boundary effects at the edge of the pack may also play an important role." We are not sure that that's the case. We believe that most of the discrepancy arises from the misuse of Eqn. 13 and the shear forces. For example, in Fig. 4a, it appears that the pack started separating at the top and fractured downward, which is why there is an X breaking pattern at the bottom.

**lines 205–206:** "On the other hand, when the forcing is more diffuse (large wavelength), the breakage is spread over a larger area, but the fracture lines are less well distinguishable." That is not what we see from Fig. 4. We observe clear fracture lines in both cases, with residual stresses in the bonds resulting from the weak velocity gradient. What we are seeing is different magnitudes of local divergence, but the fracture lines are both well-defined.

**line 208:** "ocean–ice" en dash.

**fig. 5:** Needs to be redone in light of eqn. 13.

**table 2:** It is concerning to me that the bond stiffness is that high. See above. Its units are in a different format than those in Table 1.

**fig. 7:** We don't think the floe color scheme is optimal. Can you change it?

Sincerely,
Antoine Savard and Arttu Polojärvi

---

## Referee Comment (RC2)

**Overall comments**

A very interesting paper that presents an analytic model for the breaking of an ice floe under uniaxial tension. The paper uses a bonded particle model under idealized conditions to validate the analytic model. Next, the authors use MODIS imagery to generate initial conditions to study landfast ice with the bonded particle model with ocean forcing via a QG model. They run this model for 48 days to show that the breakage properties depend upon the deformation radius and bond strength. The paper represents a significant contribution to our knowledge of the ice dynamics in the context of landfast ice, but requires a major revision. The paper also has numerous typographical errors that require correction.

**Major comments/concerns**

**1. Critical fracture force.**

The authors use a method in which the bond is broken when the absolute value of the normal force exceeds a critical value which implies there would be the same threshold for both compression and tension. The author notes that they disable compressive failures, however it is well known that compressive failures are important to ice dynamics. However as authors note ice is much stronger under compression than tension, so they will need to update equation (3) and the their breakage criteria to reflect this.

**2. Dominant Balance Assumption**

In section 3.1 the authors say for strong enough oceanic forcing, we may assume that the magnitude of the drag force is much larger than the bond force such that the bond force can be ignored in the momentum balance. The authors never give a justification for this, and I wonder how strong the ocean forcing would have to be. When the authors run the models to compare their analytic solution, the bond stiffness they use is so large that given the values they present in the tables at the strongest ocean forcing of 1m/s the bonds would stretch with a relative 1/10th of a millimeter for a 500m radius disk to have the bond and drag forces in balance with each other. For the more realistic forcing the relative displacement is order of microns.

**3. Numerical Stability**

The authors use very spring stiffnesses, especially in the eddy model. These stiffnesses are accompanied by a large timestep 10s. At these large timesteps, I worry that the authors are not fully resolving the waves propagating through their DEM.

**4. Eddy damping scheme**

Yedge is chosen by the authors to be 30km and is shown in figure 7. The authors state that the results are not sensitive to the definition of it. However figure 7 seems to show breakage in many of the figures along this line that is plotted. So which results are the author saying do not depend on this definition?

**Minor comments and suggestions**

**Abstract**

Line 4: Maybe want to indicate here or somewhere else that the LS stands for level set

Line 7: Power law exponent. Only slope when plotted on a log-log plot

**Introduction**

Line 28: You introduce FSD in abstract so not sure you need to define it again here.

Line 41: ", than waves that affect smaller scales..." should be "than by waves"

Line 54: I'm not sure geometry is the best choice of words here

Line 69: "break mode and the evolution of the pack." Missing comma "break mode, and the evolution of the pack."

**The bonded particle method for sea ice**

Fig. 2: I would consider moving figure 2b to 2c since the reference to it comes so much later. Line 95: How are you calculating ocean velocity when the centroid of the ice does not exactly match up with the ocean grid?

**Uniaxial bond force**

General comment for this section: Missing subscript i on many of these equations Eqn 7: Does gamma represent anything physically or was this just a mathematical convenience?

Eqn 8: I feel like the symbol epsilon is typically used to represent strain, where here this is a displacement. So not sure this is the best choice of a symbol in the context

Eqn 8: I know for your tests you run later the ocean is set to a specific velocity, but can we generally assume that it is independent of time like you did here?

**Breakage length scale under a pulse forcing**

Line 146: Say here that x=0 is located at the center of the bond

Eqn 12: Missing  $\frac{1}{4}$  in the  $x^c$  equation, but appears to have been accounted for eqn 13. Also feels strange that this is dependent upon t. So this integration starts from the moment the forcing is applied to move it out of unstretched neutral position?

Eqn 13: Looking at fig 3a, the two parts of the bond force greater than Fc are not continuous, but it seems here are adding them together?

Fig 3b: Something is off with this plot. Ibr should equal 0 at  $\lambda^{cut}$ . The way  $\lambda^{cut}$  is defined, eqn 13 should have an arcsin(1), which means the right hand side should equal 0.

**Setup**

Line 182: How frequently are bonds evaluated to see if they exceed the critical fracture value? This will impact  $x^c$  from your analytic solutions for validation. Any significance to 11 hours and how this value was chosen for the duration of the simulation?

Line 184: The way you say you evaluate the simulated breakage length, would this include the central 2xc value shown in figure 3a?

Table 1: include values for ice and ocean density.

**Results**

Line 188-194: what's the diffuse ocean forcing? 100km or 200km because it is not consistent between the text and the figure.

Fig 4: A video of this simulation would be nice to have in supplemental material. This will allow us to see any leads form and how fractures appear. I assume we would see the fractures appear at  $\pm \lambda/8$  first as we see this is the highest bond force in equation 11. I am curious if we see a stress wave propagate out from when the fractures happen. It is this wave of stress in spaghetti that causes it to fracture in multiple places and this is reminiscent of that process.

Fig 5: Given there seems to be issues with figure 3, check to see if analytical solutions

**Eddying current properties**

239-240: Are these run to quasi-equilibrium with the damping in place or is that added after?

**Results**

Fig 8: A video of this simulation would be nice to have in supplemental material. This will allow us to see any leads form and how fractures appear. Since we see the large changes to the ice area in 8a, it would be good to see where these losses appear.

Fig 8c-d: These are never specifically mentioned in the text though you definitely talk about them. So put these references in.

Fig 8 caption: Cite the paper with the method used for calculating FSD since there are different methods.

Line 293: What is the bin for your smallest FSD? Since we see fracturing down to the smallest grain, I worry that we might see some edge effects in the power-law you find if you include single grains since those are no longer able to fracture into smaller ones.

**Code and data availability:**

Line 340: Do you need a zenodo DOI or anything like that?